# Microbial effects of prebiotics, probiotics and synbiotics after Caesarean section or exposure to antibiotics in the first week of life: A systematic review

Nora C. Carpay[1]⊕*, Kim Kamphorst[1]⊕, Tim G. J. de Meij[2], Joost G. Daams[3], Arine M. Vlieger[4‡], Ruurd M. van Elburg[1‡]

1 Department of Paediatrics, Amsterdam UMC, Location University of Amsterdam, Amsterdam Gastroenterology and Metabolism Research Institute, Amsterdam, The Netherlands, 2 Department of Paediatric Gastroenterology, Amsterdam UMC, Location Vrije Universiteit Amsterdam, Amsterdam, The Netherlands, 3 Amsterdam UMC, Location University of Amsterdam, Medical Library, Amsterdam, The Netherlands, 4 Department of Paediatrics, St. Antonius Hospital, Nieuwegein, The Netherlands

⊕ These authors contributed equally to this work.
‡ These authors also contributed equally to this work.
* n.c.carpaij@amsterdamumc.nl

## Abstract

### Background and aims

Disruption of the developing microbiota by Caesarean birth or early exposure to antibiotics may impact long-term health outcomes, which can potentially be prevented by nutritional supplements. This systematic review aimed to summarise the evidence regarding the effects of prebiotics, probiotics and synbiotics on the intestinal microbiota composition of term infants born by Caesarean section or exposed to antibiotics in the first week of life.

### Methods

A systematic search was performed from inception to August 2022 in Medline and Embase. Two researchers independently performed title and abstract screening (n = 12,230), full-text screening (n = 46) and critical appraisal. We included randomised controlled trials which included term-born infants who were born following Caesarean section or who were exposed to postpartum antibiotics in the first week of life, pre-, pro- or synbiotics were administered <6 weeks after birth and outcome(s) consisted of microbiota analyses.

### Results

Twelve randomised controlled trials investigating Caesarean born infants and one randomised controlled trial including infants exposed to antibiotics were included. Group sizes varied from 11 to 230 with 1193 infants in total. Probiotic (n = 7) or synbiotic (n = 3) supplementation significantly increased the abundance of the supplemented bacterial species (of the *Bifidobacterium* and *Lactobacillus* genus), and there was a decrease in *Enterobacteriaceae*, especially <4 weeks of age. At phylum level, Actinobacteria (two studies),

**Data Availability Statement:** All relevant data are within the paper and its Supporting Information files.

**Funding:** The author(s) received no specific funding for this work.

**Competing interests:** The authors have declared that no competing interests exist.

Proteobacteria (one study) and Firmicutes (one study) increased after probiotic supplementation. In three studies on prebiotics, two studies reported a significant increase in *Bifidobacteria* and one study found a significant increase in *Enterobacteriaceae*.

## Discussion

Prebiotic, probiotic and synbiotic supplements seem to restore dysbiosis after Caesarean section towards a microbial signature of vaginally born infants by increasing the abundance of beneficial bacteria. However, given the variety in study products and study procedures, it is yet too early to advocate specific products in clinical settings.

## Introduction

The human gastrointestinal microbiota is a collection of all microorganisms (bacteria, viruses, fungi, protozoa) residing in the gastrointestinal tract. Together, these microorganisms affect processes such as metabolism [1,2] and inflammatory and immunological responses[2], and also influence the integrity and structure of the gastrointestinal tract [2]. The normal gastrointestinal microbiota develops rapidly after birth and is highly dynamic until it shifts towards an adult-like composition around the age of three years [3]. This development is driven by exposure to microbes from maternal, environmental, and dietary sources [4] and can be disrupted by many factors, especially when they occur early in this developmental process.

Caesarean section is one of the most important causes of disrupted microbiota development due to reduced vertical mother-infant transmission of beneficial intestinal bacteria (specifically of the *Lactobacillus* and *Bifidobacterium* genus). It has been suggested that the prenatal antibiotic exposure during a Caesarean section also affects the infant's microbiota development, but a recent randomised controlled trial (RCT) reported that prenatal exposure to antibiotics during caesarean section does not further disrupt the microbiota colonisation [5]. During and after a Caesarean section, the infant becomes predominantly colonised with bacteria from the hospital environment (e.g. *Staphylococcus*, *Corynebacterium* and *Propionibacterium* species) [1,2,6,7]. Dysbiosis after Caesarean section can persist for as long as seven years and is associated with a higher risk of obesity, atopy, and type 1 diabetes mellitus [6].

In addition to Caesarean deliveries, exposure to antibiotics in early life has been associated with dysbiosis [8]. Early life antibiotics have been shown to decrease the abundance of *Bifidobacteria* [9] and *Bacteroidetes* [10] and increase the amount of *Clostridia* [8] and *Enterobacteriaceae* [9]. Antibiotics are the most frequently prescribed drugs in neonates with 8% of all European infants exposed to antibiotics in the first week of life [11]. The effect of antibiotic exposure, specifically in the first week of life, has been associated with an altered gut microbiota[9,12] a higher risk of wheezing [13], infantile colic [13], gastrointestinal disorders [14], impaired growth [12,15], allergies [16], and asthma [17].

These short- and long-term health effects linked to early dysbiosis through Caesarean delivery and neonatal antibiotic exposure illustrate the need for interventions aimed at restoration of this dysbiosis, and consequently prevention of related health consequences. Supplementation with prebiotics, probiotics or synbiotics has been described as a promising intervention to reduce some of the risks associated with early microbiota disruption. Probiotics are live microorganisms such as *Bifidobacteria* and *Lactobacilli* [7], while prebiotics are nutrients that promote growth and activity of bacteria that already exist in the gut [18]. Synbiotics are a combination of pre- and probiotics [18].

The aim of this systematic review is to identify all studies investigating the effects of a pre-, pro- or synbiotic supplement on the gut microbiota of term-born infants born by Caesarean section or exposed to antibiotics in the first week of life.

## Methods

### Literature search

OVID Medline and Embase were systematically searched from inception to August 10, 2022. The search strategy was constructed in collaboration with a medical librarian (JD) and was composed of the following components:

([c section] OR ([antibiotic treatment] AND [first week of life] OR [first week antibiotics])) AND

- [pre- pro- synbiotics]

  OR

- [dietary supplements] AND [microbiome]

  OR

- [dietary supplements brands]

In order to reduce recall bias and enhance search results precision VOS-viewer was used to identify terms for NOTing out irrelevant records from databases searched [19]. No other filters or limits were used. The full search term including the specific keywords and combinations of search components can be found in the S1 Table.

### Eligibility criteria

The following inclusion criteria were applied, all criteria had to be met for inclusion:

1. study participants were term-born infants who were born following Caesarean section or exposed to antibiotics in the first week of life (born vaginally or following Caesarean section),

2. administration of pre-, pro- or synbiotic dietary supplements was started within six weeks after birth,

3. reported outcome(s) consisted of microbiota analyses,

4. study design was a randomised controlled trial.

   Exclusion criteria were:

1. studies including infants with major congenital malformations,

2. studies written in a language other than English,

3. animal studies,

4. for the Caesarean-analyses: studies which included both vaginally and Caesarean-delivered infants but performed no subgroup analyses for only the Caesarean-delivered infants

### Data collection

All records found in the search were exported into Rayyan after deduplication [20]. Two researchers (NC and KK) independently performed title and abstract screening, as well as full-

text screening. Titles and abstracts were screened by determining whether the article could meet the in- and exclusion criteria stated above. After consensus on the included articles, relevant data was extracted by NC in consultation with the other co-authors. Reference lists of the included articles were hand-searched to look for additional relevant studies.

All significant outcomes provided in the main text or supplemental information were summarised in a table, and non-significant results from studies investigating the same outcomes were reported in separate bar charts.

If both "per protocol" and "(modified) intention to treat" analyses were available, only the results from the "(modified) intention to treat" analysis were included in the table.

## Critical appraisal

To assess the risk of bias in the included articles, the Cochrane risk-of-bias tool for randomised controlled trials (RoB 2) [21] was used. The RoB 2 assesses the risk of bias of studies in five domains: bias arising from the randomisation process, bias due to deviations from the intended intervention, bias due to missing outcome data, bias in measurement of the outcome, and bias in selection of the reported results. Risk of bias was independently assessed by two researchers (NC and KK) and any discrepancies were discussed until a consensus was reached. The guidance document of the RoB 2 was used to determine whether articles had a high, some or a low risk of bias. If a study included both vaginally and Caesarean-delivered infants and performed a subgroup analysis on the Caesarean-delivered infants, only the methods used for the relevant subgroup analyses were assessed.

The review and protocol were not registered. This systematic review was conducted according to the guidelines of Preferred Reporting Items for Systematic Reviews and Meta-Analyses [22].

## Results

Of the 15,756 records, 12,230 remained after deduplication. After title and abstract screening, 56 articles were deemed suitable for full-text screening. Finally, 13 articles were included for analysis (Fig 1). Hand-searching the reference lists of these articles did not result in any more inclusions.

## Study characteristics

In total, 13 articles were included, based on 12 randomised controlled trials (Fig 1); Lay et al [23] published results of a subgroup analysis based on the RCT by Chua et al. [24].

The 12 microbiota studies investigated the effect of prebiotics [23–26] (n = 3), synbiotics [23,24,27,28] (n = 3) and probiotics [29–35] (n = 7) (Table 1). The interventions were started between birth and the first three weeks of life, and treatment duration varied between five days after birth and six months of age. All studies investigated the effect of these interventions on infants born via Caesarean section, except for one study which included infants who received antibiotic treatment within three days after birth [35].

## Critical appraisal

The assessment of the risk of bias of the included studies is provided in Table 2. Of the 12 studies, 10 were determined to have a high risk of bias, mainly due to issues in adhering to the intervention. Most studies did not address the extent to which participants adhered to the intervention, and if they did, the appropriate analyses necessary to estimate the effect of the non-adherence to the intervention were not applied.

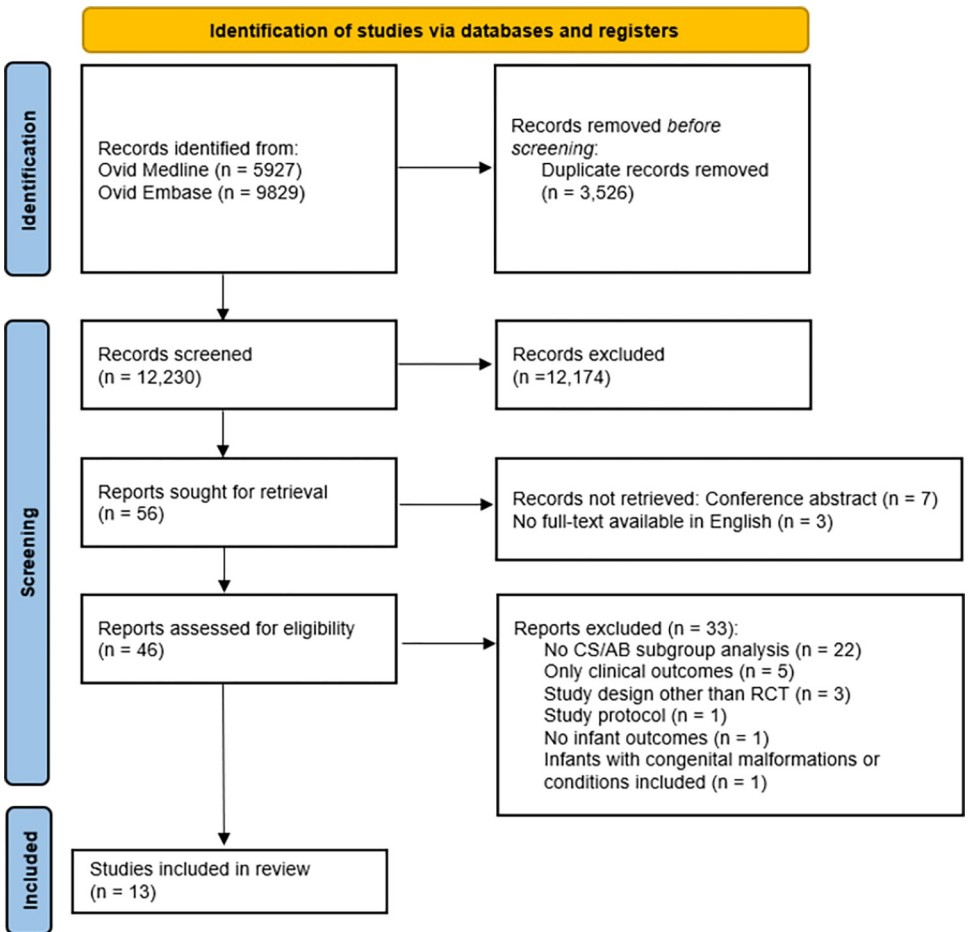

**Fig 1. Flowchart showing article selection.** Adapted from the PRISMA 2020 flow diagram [22].

## The effect of pre-, pro- and synbiotics after antibiotics in the first week of life

The effects of the interventions were divided in three time clusters: 0–1 weeks, 1–4 weeks, or >4 weeks. The only study during or after antibiotic treatment in the first week of life investigated the effect of a probiotic supplement containing *Bifidobacterium longum*, *Lactobacillus acidophilus* and *Enterococcus faecalis* [35]. At the phylum level they found a significant increase in abundance of Actinobacteria and Proteobacteria at 0–1 weeks and >4 weeks and an increase in Actinobacteria at 1–4 weeks. At the genus level, they reported a significant increase in the relative abundance of *Bifidobacterium* at 0–1 weeks and >4 weeks (Table 3).

## The effect of pre-, pro- and synbiotics after Caesarean section

Fig 2 summarises the number of statistically significant and non-significant differences that were found in the three time clusters. The significant microbiota changes in the experimental groups compared to the control groups (described in Table 1) are discussed in further detail below.

**Diversity and compositional differences.** One study [23] found compositional differences (using a distance-based redundancy analysis) at 0–1 weeks, 1–4 weeks and >4 weeks in the infants who received a synbiotic compared to those who received a prebiotic or placebo,

Table 1. General characteristics of the included randomised controlled trials.

| First author | Country | Study period (year published) | # Participants[1] | | | AB or CS SG? | Feeding method | Intervention | Control | Start of intervention | Duration intervention | Outcomes (relevant subgroup) | Follow-up | Comments |
|---|---|---|---|---|---|---|---|---|---|---|---|---|---|---|
| | | | I | C | T | | | | | | | | | |
| Chua [24] | Singapore & Thailand | 2011–2013 (2017, 2021) | 52 + 51 | 50 | 153 | CS | Mixed (FF + BF) | Prebiotic (scGOS/lcFOS) or synbiotic (scGOS/lcFOS and Bifidobacterium breve M-16V) | Control formula | 1–3 D | 16 W | Total faecal Bifidobacterium, Bifidobacterium species abundance, other members of the gut microbiota, pH, sc fatty acids, lactate | 3, 5 D 2, 4, 8, 12, 16, 22 W | Infants born via CS were also exposed to intrapartum AB prophylaxis. Lay et al.: some results were based on a SG |
| Lay [23] | | | 39 + 44 | 44 | 127 | | | | | | | | | |
| Berger [25] | Italy & Belgium | 2012–2015 (2020, 2017) | 19 | 24 | 43 | CS SG | Exclusive FF | Prebiotics: 2 HMOs (2'-fucosyllactose and lacto-N-neotetraose) | Control formula | 0–14 D | 6 M | Stool microbiota diversity | 3, 12 M | - |
| Korpela [29] | Finland | 2000–2003 (2018, 2018, 2009, 2017) | 35 | 44 | 79 | CS SG | Exclusive BF, mixed feeding or FF | Probiotic: Lactobacillus rhamnosus LC705, Bifidobacterium breve Bb99, Propionibacterium freudenreichii spp., shermanii JS | Placebo (micro-crystalline cellulose) | 36 W gestation + from birth | 6 M | Microbiota composition | 3 M | Infants had to be at risk for atopic disease (at least one parent with asthma, allergic rhinitis or eczema) and this intervention was initiated prenatally (36 W gestation) |
| Baglatzi [30] | Greece | 2009–2011 (2016) | 84 | 80 | 164 | CS | Exclusive or mixed FF | Probiotic: regular dose of Bifidobacterium lactis | Low dose of B. lactis | Birth | 6 M | Detection of B. lactis | 12 M | No control group that was fed formula without pre-, pro-, or synbiotics |
| Cooper [27] | South Africa | 2008–2013 (2016) | 92 | 101 | 193 | CS SG | Exclusive FF | Synbiotic: BMOs (containing GOS and MOS such as 3'- and 6' sialyllactose) + Bifidobacterium lactis CNCM-I-3446 | Control formula | Birth (≤3 D) | 6 M | Faecal (bifido) bacteria, anthropometrics, faecal pH, lean mass, fat mass and bone mineral content, digestive tolerance, immune parameters, HIV infection status, frequency of morbidity episodes | 1 Y | All included infants had HIV + mothers and all mothers and infants received antiretroviral medication. Infants who tested positive for HIV were excluded |

(Continued)

**Table 1.** (Continued)

| First author | Country | Study period (year published) | # Participants[1] I | C | T | AB or CS SG? | Feeding method | Intervention | Control | Start of intervention | Duration intervention | Outcomes (relevant subgroup) | Follow-up | Comments |
|---|---|---|---|---|---|---|---|---|---|---|---|---|---|---|
| Estorninos [26] | Philippines | 2016–2018 (2022) | 115 | 115 | 230 | CS SG | Exclusive FF | Prebiotic: bovine MOS (GOS and sialylated-oligosaccharides) | Control formula | 3 W | 6 M | Phylogenetic distance/microbiota composition, Bifidobacteria abundance | 4 M | At the 2.5 month time point, only a subgroup of 75 infants for each group provided a faecal sample |
| Frese [31] | USA | 2015–2016 (2017) | 11 | 9 | 20 | CS SG | Any | Probiotic: Bifidobacterium infantis EVC001 | None | 7 D | 27 D | Microbiota composition, relative abundances of the most abundant taxonomic groups | 60 D | Significantly more mothers in the control group were primiparous |
| Garcia Rodenas [32] | Greece | 2010–2011 (2016) | 11 | 10 | 21 | CS SG | Exclusive FF | Probiotic: Lactobacillus reuteri DSM 17938 | Control formula | <72 H | 6 M | Relative abundance of OTUs, weighted UniFrac distances, relative abundance of Bifidobacterium | 4 M | - |
| Hurkala [33] | Poland | 2014–2017 (2020) | 71 | 77 | 148 | CS | Exclusive FF | Probiotic: Bifidobacterium breve PB04 and Lactobacillus rhamnosus KL53A | None | <1 H | Until discharge (5 or 6 D) | Abundance of lactobacilli in faeces, populations of Bifidobacterium in faeces, populations of potentially pathogenic bacteria | 1 M | Significantly more missing stool samples from the control group (29 compared to 13 in the intervention group) |
| Roggero [34] | Italy | 2015–2016 (2020) | 16 | 16 | 32 | CS SG | Exclusive FF | Probiotic: Lactobacillus paracasei CBA L74 | Control formula | <7 D | 3 M | sIgA production, antimicrobial peptides, microbiota diversity, metabolome, abundance of bacterial genera | 90 D | Infants may have been breastfed for a few days before enrolment |
| Yang [28] | China | 2018 (2021) | 7 + 7 | 9 | 23 | CS | BF | Synbiotic: high and low dose of Bifidobacterium lactis Bi-07 and Lactobacillus rhamnosus HN001 + GOS | No probiotic | Birth | 28 D | Diversity of gut microbiota, gut microbiota composition, COGs | 28 D | - |

*(Continued)*

**Table 1.** (Continued)

| First author | Country | Study period (year published) | # Participants[1] | | | AB or CS SG? | Feeding method | Intervention | Control | Start of intervention | Duration intervention | Outcomes (relevant subgroup) | Follow-up | Comments |
|---|---|---|---|---|---|---|---|---|---|---|---|---|---|---|
| | | | I | C | T | | | | | | | | | |
| Zhong [35] | China | 2017–2018 (2021) | 25 + 13 | 17 | 55 | 1 week AB | Any | Probiotic: *Bifidobacterium longum*, *Lactobacillus acidophilus* and *Enterococcus faecalis* during or after AB treatment | None | Beginning or end of AB treatment (AB treatment started <3 D after birth) | 42 D | Gut microbiota, relative abundance of OTUs | 42 D | Children were not necessarily born via CS, but received AB in the first week of life |

[1] # participants in a subgroup, if applicable.

I Intervention.

C Control.

T Total.

CS Caesarean section.

SG subgroup.

BF breastfeeding.

FF formula feeding.

HMOs human milk oligosaccharides.

(sc) GOS: (short chain) galactooligosaccharides.

(lc) FOS: (long chain) fructooligosaccharides.

Spp. Several species.

BMOs bovine milk oligosaccharides.

MOS: Milk oligosaccharides.

D days.

M months.

W weeks.

H hours.

Y year.

AB antibiotic.

LRTI lower respiratory tract infection.

URTI upper respiratory tract infection.

OTU operational taxonomic unit.

sIgA secretory Immunoglobulin A.

COG: Clusters of orthologous groups of proteins.

**Table 2. Risk of bias of the included studies.**

| First author | Domains of the Cochrane risk-of-bias tool for randomised controlled trials (RoB-2) | | | | | |
|---|---|---|---|---|---|---|
| | Domain 1 | Domain 2 | Domain 3 | Domain 4 | Domain 5 | Total |
| Chua [24] | yellow | red | red | green | green | red |
| Lay [23] | yellow | red | green | green | green | red |
| Berger [25] | yellow | green | green | green | green | green |
| Estorninos [26] | green | red | green | green | red | red |
| Korpela [29] | green | green | green | green | green | green |
| Baglatzi [30] | green | red | green | green | green | red |
| Cooper [27] | green | red | green | red | green | red |
| Frese [31] | red | green | green | green | green | red |
| Garcia Rodenas [32] | green | green | green | green | green | green |
| Hurkala [33] | green | red | yellow | green | green | red |
| Roggero [34] | green | red | green | green | red | red |
| Yang [28] | green | green | red | green | yellow | red |
| Zhong [35] | green | red | green | green | green | red |

Green: Low risk of bias, yellow: Some risk of bias, red: High risk of bias.

Domain 1: Risk of bias arising from the randomisation process.

Domain 2: Risk of bias due to deviations from the intended interventions (*effect of adhering to intervention*).

Domain 3: Missing outcome data.

Domain 4: Risk of bias in measurement of the outcome.

Domain 5: Risk of bias in selection of the reported result.

and an increased diversity at 8 weeks using the Shannon diversity index. Researchers of two other studies [26,32] also measured compositional differences (phylogenetic distance) at 1–4 weeks [32] and >4 weeks [26] and reported a significantly different microbiota composition in infants who received a probiotic [32] or prebiotic [26] compared to a placebo.

**Phylum level.** Only one study [32] investigated the effect of a probiotic after Caesarean delivery on the phylum level. At 1–4 weeks, they found an increase in both Actinobacteria and Firmicutes in the probiotic group.

**Family level.** At 0–1 weeks, Chua et al. [24] found a significant decrease in the percentage of *Enterobacteriaceae* present in the stool of infants who received the synbiotic, but not those who received the prebiotic. Lay et al. [23], who analysed a subgroup of infants from the same study, reported an increase in relative abundance of strict anaerobes, a decrease in relative abundance of facultative anaerobes/aerobes and an increase in *Bifidobacteriaceae* in the synbiotic group.

At 1–4 weeks, Chua et al. again report a significant decrease in the percentage of *Enterobacteriaceae* in the synbiotic group, but a significant increase in the prebiotic group [24]. In a subgroup analysis by Lay et al., no significant differences were found in the prebiotic group, but an increase in strict anaerobes, decrease in facultative anaerobes/aerobes and *Clostridiaceae* and an increase in *Bifidobacteriaceae* was observed after a synbiotic supplement [23]. In line with their findings, another study[32] also found a significant increase in *Bifidobacteriaceae*. Additionally, they found a significant increase in *Lactobacillaceae*. Furthermore, they reported a significant decrease in the percentage of *Enterobacteriaceae* in their probiotic group, which is similar to Chua et al.'s findings in their synbiotic intervention group.

At >4 weeks, Chua et al. found the same results as at 1–4 weeks: a significant decrease in the percentage of *Enterobacteriaceae* in the synbiotic group, and a significant increase in the prebiotic group [24]. Moreover, in a subgroup a decrease in *Staphylococcaceae* was reported

**Table 3. Effects of pre-, pro- or synbiotic interventions on microbiota composition.**

| First author | Intervention | | | # Participants[1] | | | Analysis techniques | Time points | Outcomes: microbiota composition | | | | | |
| | Pre-/pro-/synbiotic | Start | Duration | I | C | T | | | Diversity + Compositional differences | Phylum level | Family level | Genus level | Species level | Other |
| | | | | | | | | **0–1 Week** | | | | | | |
| Cooper [27] | Symbiotic: BMOs (containing GOS and MOS such as 3'- and 6' sialyllactose) + B. lactis CNCM-I-3446 | Birth (≤3 D) | 6 M | 92 | 101 | 193 | PCR, FISH | 3 D | | | | n.s. | | |
| Chua [24] | Prebiotic (scGOS/ lcFOS) | 1–3 D | 16 W | 39 | 45 | 84 | 16S rRNA sequencing + FISH + qPCR | 3/5 D | | | | % Bifidobacteria: ↑ | | |
| | Synbiotic (scGOS/ lCFOS and B. breve M-16V) | 1–3 D | 16 W | 45 | 45 | 90 | | 3/5 D | | | % Enterobacteriaceae: ↓ | Estimated mean of total Bifidobacterium gene count: ↑ % Bifidobacteria: ↑ Bifidobacteria count: ↑ | B. breve M-16V [intervention] detected in infant: ↑ | Acetate: ↑ pH: ↓ |
| Lay [23] | Prebiotic (scGOS/ lcFOS) | 1–3 D | 16 W | 39 | 44 | 83 | Shotgun 16S rRNA sequencing of the V3-V6 region, shotgun metagenomics, metatranscriptomics and metabolomics | 3/5 D | | | | | n.s. | |
| | Synbiotic (scGOS/ lCFOS and B. breve M-16V) | 1–3 D | 16 W | 44 | 44 | 88 | | 3 D | Compositional difference | | Relative abundance of strict anaerobes**: ↑ Relative abundance of facultative anaerobes/ aerobes***: ↓ Bifidobacteriaceae: ↑ | Bifidobacterium: ↑ | Abundance of B. breve [intervention] | |
| | | | | | | | | 5 D | Compositional difference | | Relative abundance of strict anaerobes**: ↑ Relative abundance of facultative anaerobes/ aerobes***: ↓ Bifidobacteriaceae: ↑ | Bifidobacterium: ↑ Haemophilus: ↓ | Abundance of B. breve [intervention]: ↑ | |
| Yang [28] | High dose of synbiotic: B. lactis Bi-07 and L. rhamnosus HN001 + GOS | Birth | 28 D | 7 | 9 | 16 | 16S rRNA gene sequencing of the V3-V4 region + PCR | 3 D | | | | Relative abundance of Bifidobacterium: ↑ Relative abundance of Lactobacillus: ↑ | | |
| | | | | 6 | 8 | 14 | | 7 D | | | | Relative abundance of Lactobacillus: ↑ | | |
| | Low dose of synbiotic: B. lactis Bi-07 and L. rhamnosus HN001 + GOS | | | 7 | 9 | 16 | | 3 D | | | | Relative abundance of Bifidobacterium: ↑ Relative abundance of Lactobacillus: ↑ | | |
| | | | | 5 | 8 | 13 | | 7 D | | | | Relative abundance of Lactobacillus: ↑ | | |
| Hurkala [33] | Probiotic: B. breve PB04 and L. rhamnosus KL53A | <1 H | 5 or 6 D | 71 | 77 | 148 | PCR | 5/6 D | | | | Abundance of Lactobacilli: ↑ Abundance of Bifidobacterium: ↑ | L. rhamnosus [intervention]: ↑ B. breve [intervention]: ↑ | |

(Continued)

**Table 3.** (Continued)

| First author | Intervention: Pre-/pro-/synbiotic | Start | Duration | I | C | T | Analysis techniques | Time points | Diversity + Compositional differences | Phylum level | Family level | Genus level | Species level | Other |
|---|---|---|---|---|---|---|---|---|---|---|---|---|---|---|
| Zhong [35] | Probiotic: *B. longum*, *L. acidophilus* and *E. faecalis* during AB treatment | <3 D | 42 D | 25 | 17 | 42 | 16S rRNA gene sequencing of the V3-V4 region + PCR | 1 W | | Relative abundance Actinobacteria: ↑ Relative Abundance Proteobacteria: ↑ | | Relative abundance of *Bifidobacterium*: ↑ | | |
| | Probiotic: *B. longum*, *L. acidophilus* and *E. faecalis* after AB treatment | 7 D | 42 D | 13 | 17 | 30 | | 1 W | n.s. | | | | | |
| **1–4 Weeks** | | | | | | | | | | | | | | |
| Cooper [27] | Synbiotic: BMOs (containing GOS and MOS such as 3'- and 6' sialyllactose) + *B. lactis* CNCM-I-3446 | Birth (≤3 D) | 6 M | 92 | 101 | 193 | PCR, FISH | 10 D | | | | Faecal *Bifidobacterium* counts: ↑ Faecal detection rate of *Bifidobacteria*: ↑ Faecal detection rate of *Bacteroides*: ↑ | Faecal detection of *B. lactis* CNCM I-3446 [intervention]: ↑ | Mean faecal pH: ↓ |
| | | | | | | | | 1 M | | | | Faecal *Bifidobacterium* counts: ↑ Faecal detection rate of *Bifidobacteria*: ↑ Faecal detection rate of *Lactobacillus*: ↑ | Faecal detection of *B. lactis* CNCM I-3446 [intervention]: ↑ | Mean faecal pH: ↓ |
| Chua [24] | Prebiotic (scGOS/lcFOS) | 1–3 D | 16 W | 39 | 45 | 84 | 16S rRNA sequencing + FISH + qPCR | 2 W | | | % *Enterobacteriaceae*: ↑ | | | |
| | | | | | | | | 4 W | | | | | | pH: ↓ |
| | Synbiotic: (scGOS/lcFOS and *B. breve* M-16V) | | | 45 | 45 | 90 | | 2 W | | | % *Enterobacteriaceae*: ↓ | Estimated mean of total *Bifidobacterium* gene count: ↑ % *Bifidobacteria*: ↑ *Bifidobacteria* count: ↑ | *B. breve* M-16V [intervention] detected in infant: ↑ | pH: ↓ |
| | | | | | | | | 4 W | | | % *Enterobacteriaceae*: ↓ | Estimated mean of total *Bifidobacterium* gene count: ↑ % *Bifidobacteria*: ↑ *Bifidobacteria* count: ↑ | *B. breve* M-16V [intervention] detected in infant: ↑ | pH: ↓ |

(*Continued*)

**Table 3.** (Continued)

| First author | Intervention: Pre/pro/synbiotic | Start | Duration | # Participants[1]: I | C | T | Analysis techniques | Time points | Outcomes: Diversity + Compositional differences | Phylum level | Family level | Genus level | Species level | Other |
|---|---|---|---|---|---|---|---|---|---|---|---|---|---|---|
| Lay [23] | Prebiotic (scGOS/lcFOS) | 1–3 D | 16 W | 39 | 44 | 83 | Shotgun 16S rRNA sequencing of the V3-V6 region, shotgun metagenomics, metatranscriptomics and metabolomics | 2 + 4 W | n.s. | | | | | |
| | Synbiotic (scGOS/lcFOS and B. breve M-16V) | | | 44 | 44 | 88 | | 2 W | Compositional difference | | Relative abundance of strict anaerobes**: ↑ Relative abundance of facultative anaerobes/aerobes***: ↓ Bifidobacteriaceae: ↑ | Bifidobacterium: ↑ | Abundance of B. breve [intervention]: ↑ | |
| | | | | | | | | 4 W | | | Clostridiaceae: ↓ | | Abundance of B. breve [intervention]: ↑ | |
| Garcia Rodenas [32] | Probiotic: L. reuteri DSM 17938 | <72 H | 6 M | 11 | 10 | 21 | 16S rRNA gene sequencing of the V1-V3 regions + PCR | 2 W | Compositional difference | Relative abundance of Actinobacteria: ↑ Relative abundance of Firmicutes: ↑ | Relative abundance of Enterobacteriaceae: ↓ Relative abundance of Bifidobacteriaceae: ↑ Relative abundance of Lactobacillaceae: ↑ | Detectable Bifidobacterium: ↑ Relative abundance of Lactobacillus: ↑ | Abundance of L. reuteri [intervention]: ↑ | |
| Zhong [35] | Probiotic: B. longum, L. acidophilus and E. faecalis during AB treatment | <3 D | 42 D | 25 | 17 | 42 | 16S rRNA gene sequencing of the V3-V4 region + PCR | 2 W | n.s. | | | | | |
| | Probiotic: B. longum, L. acidophilus and E. faecalis after AB treatment | 7 D | 42 D | 13 | 17 | 30 | | 2 W | | Relative abundance of Actinobacteria: ↑ | | | | |
| Hurkala [33] | Probiotic: B. breve PB04 and L. rhamnosus KL53A | <1 H | 5 or 6 D | 58 | 48 | 106 | PCR | 1 M | | | | Abundance of Lactobacilli: ↑ | | |
| Yang [28] | High dose of synbiotic: B. lactis Bi-07 and L. rhamnosus HN001 + GOS | Birth | 28 D | 6 | 5 | 11 | 16S rRNA gene sequencing of the V3-V4 region + PCR | 1 M | n.s. | | | | | |
| | Low dose of synbiotic: B. lactis Bi-07 and L. rhamnosus HN001 + GOS | | | 7 | 5 | 12 | | | n.s. | | | | | |
| **>4 weeks** | | | | | | | | | | | | | | |
| Zhong [35] | Probiotic: B. longum, L. acidophilus and E. faecalis during AB treatment | <3 D | 42 D | 25 | 17 | 42 | 16S rRNA gene sequencing of the V3-V4 region + PCR | 42 D | | Relative abundance of Actinobacteria: ↑ Relative abundance of Proteobacteria: ↑ | | Relative abundance of Bifidobacterium: ↑ | | |
| | Probiotic: B. longum, L. acidophilus and E. faecalis after AB treatment | 7 D | 42 D | 13 | 17 | 30 | | 42 D | | | n.s. | | | |

(Continued)

**Table 3.** (Continued)

| First author | Intervention Pre/pro-/synbiotic | Start | Duration | # Participants[1] I | C | T | Analysis techniques | Time points | Diversity + Compositional differences | Phylum level | Family level | Genus level | Species level | Other |
|---|---|---|---|---|---|---|---|---|---|---|---|---|---|---|
| Chua [24] | Prebiotic (scGOS/ lcFOS) | 1–3 D | 16 W | 39 | 45 | 84 | 16S rRNA sequencing + FISH + qPCR | 12 W | | | n.s. | | | |
| | | | | | | | | 16 W | | | % Enterobacteriaceae: ↑ | | | |
| | Synbiotic (scGOS/ lcFOS and B. breve M-16V) | | | 45 | 45 | 90 | | 12 W | | | % Enterobacteriaceae: ↓ | Bifidobacteria count: ↑ | B. breve M-16V detected in infant: ↑ | |
| | | | | | | | | 16 W | | | | | B. breve M-16V detected in infant: ↑ | |
| Lay [23] | Prebiotic (scGOS/ lcFOS) | 1–3 D | 16 W | 39 | 44 | 83 | Shotgun 16S rRNA sequencing of the V3-V6 region, shotgun metagenomics, metatranscriptomics and metabolomics | 8 W | | | Staphylococcaceae: ↓ | | | |
| | | | | 44 | 44 | 88 | | 12, 16, 22 W | | | n.s. | | | |
| | Synbiotic (scGOS/ ICFOS and B. breve M-16V) | | | 10 | 10 | 20 | | 8 W | Shannon diversity: ↑ | | | | B. longum: ↓ | |
| | | | | | | | | 12, 16 W | | | n.s. | | | |
| | | | | | | | | 22 W | | | | | V. dispar: ↑ | |
| Estorninos [26] | Prebiotic: bovine MOS (GOS and sialylated-oligosaccharides) | 3 W | 6 M | 75 | 75 | 150 | 16S rRNA gene sequencing of the V3-V4 region | 2.5 M | Compositional difference | | | | | |
| | | | | 114 | 112 | 226 | | 4 M | Compositional difference | | | Abundance of Bifidobacterium: ↑ | | |
| Berger [25] | Prebiotics: 2 HMOs (2'-fucosyllactose and lacto-N-neotetraose) | 0–14 D | 6 M | 19 | 24 | 43 | 16S rRNA gene sequencing of the V3-V4 region | 3 M | | | n.s. | | | |
| Korpela [29] | Probiotic: L. rhamnosus LC705, B. breve Bb99, P. freudenreichii spp., shermanii JS and GOS | 36 W gest. + from birth | 6 M | 35 | 44 | 79 | 16S rRNA gene amplicon sequencing of the V3-V4 region | 3 M | | | Bifidobacteriaceae: ↑ Coriobacteriaceae: ↑ Porphyromonadaceae: ↑ Bacteroidaceae: ↑ | | | |
| Cooper [27] | Synbiotic: BMOs (containing GOS and MOS such as 3'- and 6' sialyllactose) + B. lactis CNCM-I-3446 | Birth (≤3 D) | 6 M | 92 | 101 | 193 | PCR, FISH | 3 M | | | | Faecal Bifidobacteria counts: ↑ Faecal detection rate of Clostridium/ Eubacterieum: ↓ | Faecal detection of B. lactis CNCM I-3446 [intervention]: ↑ | Mean faecal pH: ↓ |
| Roggero [34] | Probiotic: L. paracasei CBA L74 | <7 D | 3 M | 16 | 16 | 32 | 16s RNA gene sequencing of the V3 region | 3 M | | | | | | sIgA production: ↑ |
| Baglatzi [30] | Probiotic: regular dose of B. lactis | Birth | 6 M | 84 | 80 | 164 | PCR | 4 M | | | | | Positive detection of B. lactis [intervention]: ↑ | |

(Continued)

**Table 3.** (Continued)

| First author | Intervention | | | # Participants[1] | | | Analysis techniques | Time points | Outcomes: microbiota composition | | | | | |
| | Pre-/pro-/synbiotic | Start | Duration | I | C | T | | | Diversity + Compositional differences | Phylum level | Family level | Genus level | Species level | Other |
|---|---|---|---|---|---|---|---|---|---|---|---|---|---|---|
| Garcia Rodenas [32] | Probiotic: *L. reuteri* DSM 17938 | <72 H | 6 M | 11 | 10 | 21 | 16S rRNA gene sequencing of the V1-V3 regions + PCR | 4 M | | | | | Abundance of *L. reuteri* [intervention]: ↑ | |

*: g. bifidobacterium, o. pseudomonadales, f. actinomycetaceae, k. bacteria, g. staphylococcus, g. streptococcus, f. streptococcaceae, f. bifidobacteriaceae, o. enterobacteriales, f. Enterobacteriaceae.

**: f. Prevotellaceae, f. peptostreptococcaceae, f. ruminococcaceae, o. clostridiales, f. porphyromonadaceae, f. clostridiaceae, f. lachnospiraceae, f. veillonellaceae, f. bacteroidaceae, f. bifidobacteriaceae.

***: o. lactobacillales, o. bacillales, f. pasteurellaceae, f. staphylococcaceae, f. lactobacillaceae, f. enterococcaceae, o. enterobacteriales, f. streptococcaceae, f. Enterobacteriaceae.

[1] # participants in a subgroup, if applicable.

I Intervention.

C Control.

T Total.

CS Caesarean section.

SG subgroup.

HMOs human milk oligosaccharides.

(sc) GOS: (short chain) galactooligosaccharides.

(lc) FOS: (long chain) fructooligosaccharides.

BMOs bovine milk oligosaccharides.

MOS: Milk oligosaccharides.

D days.

M months.

W weeks.

H hours.

Y year.

AB antibiotics.

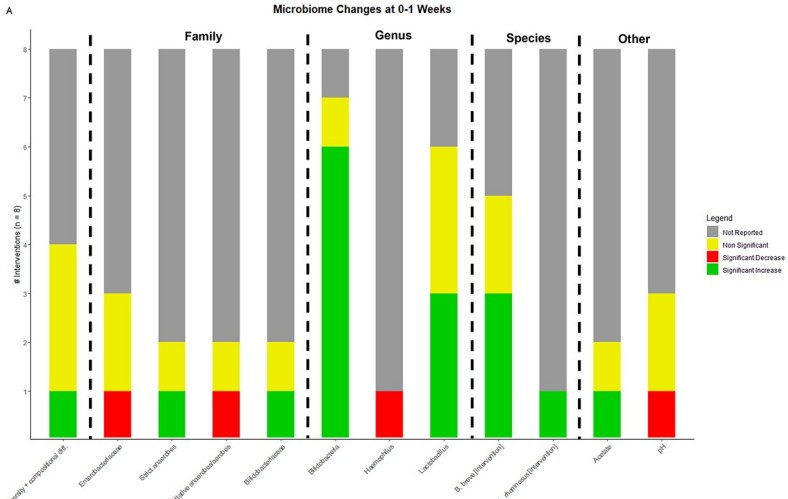

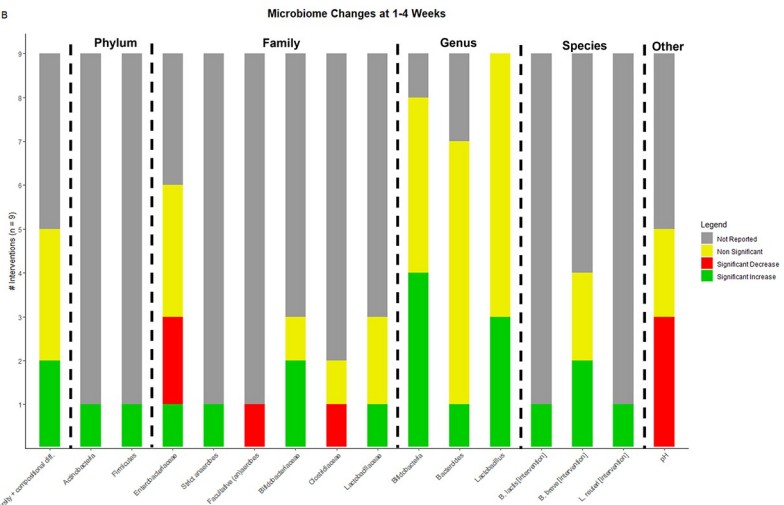

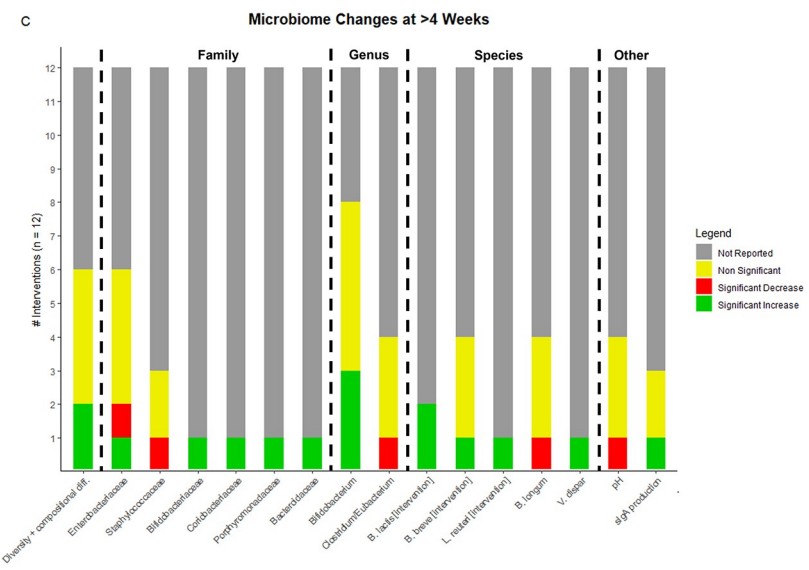

**Fig 2.** Bar charts showing the number of interventions (several studies used more than one intervention) with a significant effect on the microbiota composition at at least one time point in the clusters of 0–1 weeks (a), 1–4 weeks (b) or >4 weeks (c).

[23]. Another study found an increase in *Bifidobacteriaceae*, *Coriobacteriaceae*, *Porphyromona-daceae* and *Bacteroidaceae* [29].

**Genus level.** At 0–1 weeks, four articles [23,24,28,33] based on three RCTs reported a significant increase in abundance of the *Bifidobacterium* genus after administration of a probiotic[33], prebiotic [24], and synbiotic [23,24,28]. Two studies [28,33] also found an increase abundance of the *Lactobacillus* genus [28,33], and one study [23] reported a decrease of the *Haemophilus* genus.

At 1–4 weeks, four articles [23,24,27,32] from three RCTs found an increase in the *Bifidobacterium* genus after administration of a probiotic [27,32] or synbiotic [23,24]. Three [27,32,33] reported an increased (relative) abundance of *Lactobacillus* and one of these [27] also observed an increased abundance of *Bacteroides*.

At >4 weeks, three studies [23,26,27] reported an increased *Bifidobacterium* genus abundance, and one of the two [27] also found a decreased faecal detection rate of *Clostridium/Eubacterium*.

**Species level.** At 0–1 weeks, Chua et al. [24] and Lay et al. [23] (based on the same RCT) reported a significant increase of *Bifidobacterium breve* M-16V detected in the infants who received a synbiotic, which included this *Bifidobacterium* species.

At 1–4 weeks, three studies [24,27,32] found an increase in the faecal detection of the bacterial species they included in their intervention: *Bifidobacterium lactis* CNCM I-3446 [27], *Bifidobacterium breve* [23] and *Lactobacillus reuteri* [32].

At >4 weeks, three studies again reported an increase in faecal detection of their intervention: *Bifidobacterium lactis* [27,30] and *Lactobacillus reuteri* [32]. Another study also found a decreased abundance of *Bifidobacterium longum* and an increase in *Veillonella dispar* [23].

**Intestinal microenvironment.** At 0–1 weeks, one article [24] found a decreased faecal pH and increased acetate after administration of a synbiotic. At 1–4 weeks, the same study [24] and another [27] both reported a decreased faecal pH after administration of a synbiotic [24,27] or prebiotic [24]. At >4 weeks, one of the studies [27] still found a decreased faecal pH in the synbiotic-group [27], and another article [34] observed an increased secretory IgA (sIgA) production in infants who received a probiotic.

## Discussion

The aim of this systematic review was to describe the effects of a pre-, pro- or synbiotic supplement on the gut microbiota following Caesarean section or exposure to antibiotics in the first week of life. Only one article investigated the effect of a probiotic on antibiotic-exposed infants; a mixture of three probiotics resulted in an increase in Actinobacteria, Proteobacteria and *Bifidobacterium*. For the Caesarean-born infants, the key finding was an increase in the supplemented bacterial species (of the *Bifidobacterium* and *Lactobacillus genus*) after probiotic or synbiotic supplements, and a decrease in *Enterobacteriaceae* after synbiotic but an increase after prebiotic supplementation. Furthermore, there were significant increases in Actinobacteria, Proteobacteria and Firmicutes in the probiotic groups compared to the control groups. Moreover, the microbiota composition of the probiotic or synbiotic group was significantly different from the control group in two studies, and bacterial species diversity was increased in one study after administration of a synbiotic.

Prebiotics are less extensively studied, and only few outcome parameters reached statistical significance. However, according to one included study, prebiotics increased the abundance of

*Enterobacteriaceae*, which has been associated with potentially negative health effects such as an increased risk of atopic eczema [36], food allergy [37] and delayed colonisation of beneficial bacterial species [36]. Three articles based on two prebiotic studies reported an increase in *Bifidobacteria* [23,24,26] and two of the three also found a significantly different microbiota composition [23,26].

Because of the heterogeneity in the interventions in terms of study design and composition of the supplement, it is difficult to compare their efficacy. Generally, probiotics and synbiotics seem more effective in increasing the abundance of beneficial bacteria. *Bifidobacteria* and *Lactobacilli*, which were increased in the intervention groups of eight and five studies respectively, are associated with various health effects: both *Bifidobacteria* and *Lactobacilli* seem to protect from allergies [38,39] and infantile colic [38] and they are associated with healthy microbiota development [38]. Several species of the *Bifidobacterium* genus are commonly present in the infant gut, and their function is to digest sugars in human milk, reduce intestinal pH and improve the integrity of the intestinal wall [40]. Delivery via Caesarean section, which was the case in five [23,24,26,28,33] of the six studies investigating the microbiota at the genus level, results in a disrupted vertical transmission of *Bifidobacterium* [40]. The results in Table 3 indicate that a pro- or synbiotic intervention can alleviate this disruption and shift the neonatal microbiota composition towards that of vaginally born infants. Human milk oligosaccharides present in breast milk can also stimulate colonisation by *Bifidobacteria* [40]. Interestingly, all five articles that reported a significant increase of *Bifidobacterium* levels at 0–1 weeks included infants that were (also) breastfed. However, at the time points after this first week, other studies that included infants who were exclusively formula fed also show significant increases in *Bifidobacterium* levels. Moreover, three studies on pro- and synbiotics also found a significantly different microbiota composition or a more diverse microbiota in the intervention groups. Birth following Caesarean section or exposure to antibiotics in the first week of life reduces bacterial diversity, which makes these infants susceptible to colonisation by bacteria usually found on the mother's skin such as *Staphylococcus*, *Corynebacterium* and *Propionibacterium* spp., which is associated with an increased risk of gastrointestinal and systemic disorders, including eczema allergies, later in life [41]. The results in Table 3 show that this diversity may (partially) be restored by supplementation with pro- or synbiotics and possibly prebiotics.

To our knowledge, this is the first systematic review evaluating the effects of pre-, pro- and synbiotics specifically in both Caesarean born and antibiotic-exposed infants. While another systematic review about the effects of pre-, pro- and synbiotics on the microbiota of children born via Caesarean section was published recently [42], we identified four additional relevant articles that were not included by Martin-Pelaez et al. Furthermore, some of their included articles did not perform a separate subgroup-analysis for children born via Caesarean section.

It is crucial to analyse these Caesarean born infants separately from vaginally born infants who were not exposed to antibiotics, because the effect of pre-, pro- and synbiotics may differ in infants with a disrupted microbiota from those who were born vaginally. Illustratively, in one of the trials included in our review, the effects of pre-, pro- and synbiotics on the microbiota was only significant in the Caesarean born subgroup [32]. Additionally, Frese et al. [31] did not perform a statistical subgroup analysis for the Caesarean born infants but the differences in the microbiota composition of their cohort seems to be largely driven by Caesarean born infants. Specifically, in their intervention group, they found a significant increase in faecal *Bifidobacteriaceae* and *Bifidobacterium infantis*, and a clear decrease in the relative abundances of *Enterobacteriaceae*, *Clostridiaceae*, *Erysipelotrichaceae*, *Pasteurellaceae*, *Micrococcaceae* and *Lachnospiraceae*. These findings suggest that especially infants with a disrupted microbiota might benefit most from an intervention with pre-, pro or synbiotics.

Important strengths of this review are the elaborate search strategy developed in collaboration with a medical librarian to include all relevant articles. We also looked for any subgroup analyses of Caesarean-born infants in the full texts, even when the title or abstract did not explicitly state that these were performed. We only included articles that performed analyses on Caesarean-born infants, and not articles that only analysed the total group of participants with vaginally born infants included.

Limitations of this study are that many articles that included a subgroup analysis of Caesarean-born infants reported only a selection of the outcomes for this subgroup. It is unclear whether more analyses were performed and only the significant results were published, which would result in publication bias. Similarly, many articles did not adjust for multiple testing. This is also reflected in the critical appraisal of the articles, which showed that 11 of the 13 included studies had a high risk of bias. Lastly, while some trials mentioned in this review included a reference group with vaginally born and/or exclusively breastfed infants, we chose to focus on analyses between intervention and placebo-controlled groups instead of also comparing the intervention groups to the reference group. For further research, it would be interesting to evaluate whether pre-, pro- and synbiotic interventions could restore the microbiota of infants born through Caesarean section or infants exposed to antibiotics to that of a healthy reference group.

Other recommendations for future research are firstly that, in order to be able to compare the results of different studies, studies should standardise their methods of faecal sample collection, storage, isolation and analyses. Furthermore, microbiota studies should increase their follow-up time to see whether any differences that were found between intervention and control groups persist beyond the duration of the intervention and to search for associations with long-term health outcomes. In addition, since the effect of a pre-, pro- or synbiotic on infants after antibiotic treatment in the first week of life was only investigated by one study [35], more RCTs are necessary in this group of infants. Moreover, to assess the clinical potential of pre-, pro- or synbiotic supplementation, it is crucial that high quality RCTs with predetermined clinical outcomes are conducted. Lastly, only one study [34] explored the effects of their intervention on the metabolome. The metabolome gives an indication of the function of the microbiota, and how the microbiota affects metabolites in urine, faeces and blood serum [43]. While most included studies focused their microbiota analysis on the microbiota composition, the metabolome might reveal important information on the mechanics by which the microbiota influences its host.

## Conclusion

Supplementation of pre-, pro- or synbiotics in Caesarean-born infants and infants who received antibiotics early in life mostly increased the phyla, families, genera and species that corresponded to the pro- or synbiotic intervention that was administered, while the effects of a prebiotic generally did not reach statistical significance. Supplementation of these at-risk children to restore the microbiota to a composition more similar to vaginally born infants (i.e. predominant colonisation by *Bifidobacteria* and *Lactobacilli*) may alleviate some of the negative consequences of a disrupted microbiota. However, more high-quality research is needed to explicate the clinical effects of such microbiota changes and to determine which pre-, pro- or synbiotic products are most effective.

## Supporting information

**S1 Table. Full search strategy.**
(DOCX)

**S2 Table. PRISMA checklist.**
(DOCX)

## Author Contributions

**Conceptualization:** Arine M. Vlieger, Ruurd M. van Elburg.

**Investigation:** Nora C. Carpay, Kim Kamphorst.

**Methodology:** Nora C. Carpay, Kim Kamphorst, Tim G. J. de Meij, Joost G. Daams, Arine M. Vlieger, Ruurd M. van Elburg.

**Supervision:** Tim G. J. de Meij, Arine M. Vlieger, Ruurd M. van Elburg.

**Visualization:** Nora C. Carpay.

**Writing – original draft:** Nora C. Carpay.

**Writing – review & editing:** Nora C. Carpay, Kim Kamphorst, Tim G. J. de Meij, Arine M. Vlieger, Ruurd M. van Elburg.

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
