## [Decision Letter · Decision Letter 0]

4 Aug 2022

PONE-D-22-18775Microbial effects of supplemented prebiotics, probiotics and synbiotics after Caesarean section or exposure to antibiotics in the first week of life: a systematic reviewPLOS ONE

Dear Dr. Carpay,

Thank you for submitting your manuscript to PLOS ONE. After careful consideration, we feel that it has merit but does not fully meet PLOS ONE’s publication criteria as it currently stands. Therefore, we invite you to submit a revised version of the manuscript that addresses the points raised during the review process.

We look forward to receiving your revised manuscript.

Kind regards,

Ozra Tabatabaei-Malazy

Academic Editor

PLOS ONE

Journal Requirements:

2. We note that this manuscript is a systematic review or meta-analysis; our author guidelines therefore require that you use PRISMA guidance to help improve reporting quality of this type of study. Please upload copies of the completed PRISMA checklist as Supporting Information with a file name “PRISMA checklist”.

Additional Editor Comments:

Reviewer#1:

In the current manuscript, the authors summarize the evidence regarding the effects of prebiotics, probiotics, and symbiotics on the intestinal microbiota composition of term infants born by Caesarean section or exposed to antibiotics in the first week of life. The authors have focused on an important and interesting clinical question and covered elements that are necessary for a systematic review; the patient, group of patients, or problem being evaluated; the intervention; comparison interventions; and specific outcomes. The discussion section provides appropriate clarification from the authors whenever it is suspected. Authors have commented on the limitations of the review itself, including those of the included studies. Authors have considered fatal flaw(s) and performed risk of bias assessments. I think the current manuscript is acceptable.

Reviewer#2:

The topic of the systematic review is very interesting and useful for the scientific community. However, authors present a review that in my opinion needs to be worked in depth. I am concern specifically on important errors in methodology and discussion. Some of the most important issues I consider should be corrected for future applications are:

1. Authors included studies until August 2021, which is almost one year ago. I would recommend to the authors to check for the new articles that fill their selection criteria published during the last year.

2. The searching strings are not clear. Authors don´t indicate which are their keywords for their searching. Neither do they present the combinations of this keywords. A first consequence of this lack of accuracy in the searching strategy is the vast number of articles obtained (11248), leading to an unnecessary amount of screening work (title and abstract). The use of filters would have also helped them.

3. Besides the high amount of information give in the 2 tables, these results have been very poorly discussed in the discussion section. The results obtained must be always discussed in the discussion section. If those are not discussed, it does not make any sense to present them.

Other observations are:

Abstract: Selection criteria should be presented in the abstract. It is important to state the design of the studies selected for the review. Bifidobacterium and Lactobacillus are not species. Please define them or re-write the sentence. When reporting variations in gut microbial populations, please indicate the group of comparison.

Introduction: Why are authors naming “microbiome” to the “collection of microorganisms”? I am aware that many researchers use the term “gut microbiota” and “gut microbiome” indistinctly, but in essence, these terms don´t mean the same. Please explain.

The lack of vertical mother-to-infant transmission is not only the possible cause of microbial dysbiosis in the newborn. Please complete.

When authors write about antibiotic exposure, they seem to present this as an issue independent of caesarean section. However, they should bear in mind that antibiotic administration to the mother also occurs during caesarian section.

The definitions of pre, pro and synbiotics should be presented.

Results

Authors should indicate the reasons of the elimination of the articles in the first step of the screening (11193).

Table 1: The interventions must be given in detail regarding the CFU of the probiotics (alone or in combination with prebiotics in symbiotic preparations) the bacterial strains used in the probiotics and in the synbiotic formulations, the prebiotic amount provided per dose, the dairy dose, and the administration route.

Footnote of table one present two times “I”, for “Intervention” and for OUT.

Critical appraisal: authors should indicate what each color means.

Please check for mistakes in the bacterial nomenclature of bacteria (specially the use of italics) throughout the manuscript, including the tables.

Authors state that “this is the first systematic review evaluating the effects of pre-, pro- and synbiotics specifically in both Caesarean born and antibiotic-exposed infants”. However only 1 out of the 12 studies are actually dealing with antibiotics.

Reviewr#3:

Dear Authors,

The manuscript is in interesting filed. However, there are some major concerns.

Overall, whole manuscript is required to English language editing.

Then, please consider following comments.

Title:

-It is better to brief.

Abstract:

-Methods section: Please insert inclusion/exclusion criteria, main keywords, update the search, and insert grey literature

Introduction:

-Nearly, more than half of Ref. are addressed in this section, please reduce Ref.

-Please address past systematic reviews and then the novelty of the current study.

Methods:

-It is better to update search, insert grey literature, and confirm the method is based on PRISMA.

-Did you search ref. lists? It is essential to hand search Ref. list of the included studies when conducting a systematic review.

-In eligibility criteria, clarify characteristics of control group, they should receive which one? prebiotics, placebo, or none of them?

-The second inclusion criteria “six weeks after birth” is ambiguous. In table 1 is observed using intervention more than 6w. What is your reason?

-In critical appraisal; the last sentence “Several studies included…..”should move to discussion.

-Please delete “the review and protocol were not registrated”.

Results:

-First line: correct dictation “deduplication”.

-“ Lay et al.[33] published results of a subgroup analysis based on the RCT by Chua et al.[34]” Why included this study? It is a duplication publication. Please describe.

-Zhong et al study is included vaginal/cesarean delivery. Please recheck the data with original paper and correct them in your tables.

Discussion:

-This section needs to revise by adding more details and discussion about your results.

Best Regards,

Eic#:

Dear Authors,

Please consider following important points:

-Prepare a response letter point by point to each comment and highlight all corrections and changes by yellow color in whole manuscript.

-Consider Language Editing.

Best Regards,

Reviewers' comments:

Reviewer's Responses to Questions

**Comments to the Author**

1. Is the manuscript technically sound, and do the data support the conclusions?

Reviewer #1: Yes

Reviewer #2: Partly

Reviewer #3: Yes

2. Has the statistical analysis been performed appropriately and rigorously? 

Reviewer #1: Yes

Reviewer #2: N/A

Reviewer #3: Yes

3. Have the authors made all data underlying the findings in their manuscript fully available?

Reviewer #1: Yes

Reviewer #2: Yes

Reviewer #3: Yes

4. Is the manuscript presented in an intelligible fashion and written in standard English?

Reviewer #1: Yes

Reviewer #2: Yes

Reviewer #3: Yes

5. Review Comments to the Author

Reviewer #1: In the current manuscript, the authors summarize the evidence regarding the effects of prebiotics, probiotics, and symbiotics on the intestinal microbiota composition of term infants born by Caesarean section or exposed to antibiotics in the first week of life. The authors have focused on an important and interesting clinical question and covered elements that are necessary for a systematic review; the patient, group of patients, or problem being evaluated; the intervention; comparison interventions; and specific outcomes. The discussion section provides appropriate clarification from the authors whenever it is suspected. Authors have commented on the limitations of the review itself, including those of the included studies. Authors have considered fatal flaw(s) and performed risk of bias assessments. I think the current manuscript is acceptable.

Reviewer #2: The topic of the systematic review is very interesting and useful for the scientific community. However, authors present a review that in my opinion needs to be worked in depth. I am concern specifically on important errors in methodology and discussion. Some of the most important issues I consider should be corrected for future applications are:

1. Authors included studies until August 2021, which is almost one year ago. I would recommend to the authors to check for the new articles that fill their selection criteria published during the last year.

2. The searching strings are not clear. Authors don´t indicate which are their keywords for their searching. Neither do they present the combinations of this keywords. A first consequence of this lack of accuracy in the searching strategy is the vast number of articles obtained (11248), leading to an unnecessary amount of screening work (title and abstract). The use of filters would have also helped them.

3. Besides the high amount of information give in the 2 tables, these results have been very poorly discussed in the discussion section. The results obtained must be always discussed in the discussion section. If those are not discussed, it does not make any sense to present them.

Other observations are:

Abstract: Selection criteria should be presented in the abstract. It is important to state the design of the studies selected for the review. Bifidobacterium and Lactobacillus are not species. Please define them or re-write the sentence. When reporting variations in gut microbial populations, please indicate the group of comparison.

Introduction: Why are authors naming “microbiome” to the “collection of microorganisms”? I am aware that many researchers use the term “gut microbiota” and “gut microbiome” indistinctly, but in essence, these terms don´t mean the same. Please explain.

The lack of vertical mother-to-infant transmission is not only the possible cause of microbial dysbiosis in the newborn. Please complete.

When authors write about antibiotic exposure, they seem to present this as an issue independent of caesarean section. However, they should bear in mind that antibiotic administration to the mother also occurs during caesarian section.

The definitions of pre, pro and synbiotics should be presented.

Results

Authors should indicate the reasons of the elimination of the articles in the first step of the screening (11193).

Table 1: The interventions must be given in detail regarding the CFU of the probiotics (alone or in combination with prebiotics in symbiotic preparations) the bacterial strains used in the probiotics and in the synbiotic formulations, the prebiotic amount provided per dose, the dairy dose, and the administration route.

Footnote of table one present two times “I”, for “Intervention” and for OUT.

Critical appraisal: authors should indicate what each color means.

Please check for mistakes in the bacterial nomenclature of bacteria (specially the use of italics) throughout the manuscript, including the tables.

Authors state that “this is the first systematic review evaluating the effects of pre-, pro- and synbiotics specifically in both Caesarean born and antibiotic-exposed infants”. However only 1 out of the 12 studies are actually dealing with antibiotics.

Reviewer #3: Dear Authors,

The manuscript is in interesting filed. However, there are some major concerns.

Overall, whole manuscript is required to English language editing.

Then, please consider following comments.

Title:

-It is better to brief.

Abstract:

-Methods section: Please insert inclusion/exclusion criteria, main keywords, update the search, and insert grey literature

Introduction:

-Nearly, more than half of Ref. are addressed in this section, please reduce Ref.

-Please address past systematic reviews and then the novelty of the current study.

Methods:

-It is better to update search, insert grey literature, and confirm the method is based on PRISMA.

-Did you search ref. lists? It is essential to hand search Ref. list of the included studies when conducting a systematic review.

-In eligibility criteria, clarify characteristics of control group, they should receive which one? prebiotics, placebo, or none of them?

-The second inclusion criteria “six weeks after birth” is ambiguous. In table 1 is observed using intervention more than 6w. What is your reason?

-In critical appraisal; the last sentence “Several studies included…..”should move to discussion.

-Please delete “the review and protocol were not registrated”.

Results:

-First line: correct dictation “deduplication”.

-“ Lay et al.[33] published results of a subgroup analysis based on the RCT by Chua et al.[34]” Why included this study? It is a duplication publication. Please describe.

-Zhong et al study is included vaginal/cesarean delivery. Please recheck the data with original paper and correct them in your tables.

Discussion:

-This section needs to revise by adding more details and discussion about your results.

Best Regards,

6. PLOS authors have the option to publish the peer review history of their article (what does this mean?). If published, this will include your full peer review and any attached files.

Reviewer #1: **Yes: **Solaleh Emamgholipour

Reviewer #2: No

Reviewer #3: No

---

## [Author Response · Author response to Decision Letter 0]

18 Sep 2022

Amsterdam, September 2022

PLOS ONE

Dr. Tabatabaei-Malazy

Dear Dr. Tabatabaei-Malazy,

We would like to thank you for the opportunity to revise and resubmit our manuscript “Microbial effects of prebiotics, probiotics and synbiotics after Caesarean section or exposure to antibiotics in the first week of life: a systematic review” for publication in PLOS ONE.

The thorough feedback of the reviewers is very much appreciated, and we have incorporated their comments to further improve our systematic review. On the following pages, we reply to each comment in chronological order. We hope that the revised manuscript meets the standards for publication in PLOS ONE.

We look forward to hearing from you regarding our submission, and we would be happy to address any further comments.

With kind regards,

on behalf of all authors,

Nora Carpay

First of all, we would like to thank the reviewers for their critical view and constructive remarks. We will reply to their comments in chronological order.

Reviewer #1:

In the current manuscript, the authors summarize the evidence regarding the effects of prebiotics, probiotics, and symbiotics on the intestinal microbiota composition of term infants born by Caesarean section or exposed to antibiotics in the first week of life. The authors have focused on an important and interesting clinical question and covered elements that are necessary for a systematic review; the patient, group of patients, or problem being evaluated; the intervention; comparison interventions; and specific outcomes. The discussion section provides appropriate clarification from the authors whenever it is suspected. Authors have commented on the limitations of the review itself, including those of the included studies. Authors have considered fatal flaw(s) and performed risk of bias assessments. I think the current manuscript is acceptable.

We thank the reviewer for their positive assessment.

Reviewer #2:

The topic of the systematic review is very interesting and useful for the scientific community. However, authors present a review that in my opinion needs to be worked in depth. I am concern specifically on important errors in methodology and discussion. Some of the most important issues I consider should be corrected for future applications are:

1. Authors included studies until August 2021, which is almost one year ago. I would recommend to the authors to check for the new articles that fill their selection criteria published during the last year.

We agree that the original search was performed quite some time ago. We repeated the search on August 10 2022 and included one more article by Estorninos et al., which was published in January 2022.

2. The searching strings are not clear. Authors don´t indicate which are their keywords for their searching. Neither do they present the combinations of this keywords. A first consequence of this lack of accuracy in the searching strategy is the vast number of articles obtained (11248), leading to an unnecessary amount of screening work (title and abstract). The use of filters would have also helped them.

We have clarified our search string by adding that the complete search string including combinations of keywords can be found in the first supplemental table (S1 table), not in the main text of the review. Indeed, we have obtained a high number of articles by using this search string. However, we do not think that this was a lack of accuracy. We did not want to be too specific in our search string to avoid excluding any relevant articles and we constructed this search string in collaboration with a medical librarian. We did use a VOS-viewer to exclude some irrelevant keywords to narrow down the articles we found. Furthermore, we made sure that two independent reviewers carefully screened the titles and abstracts to make sure no relevant articles were missed, even though there was a high number of articles to be screened.

3. Besides the high amount of information give in the 2 tables, these results have been very poorly discussed in the discussion section. The results obtained must be always discussed in the discussion section. If those are not discussed, it does not make any sense to present them.

We agree that there are several results presented in the tables which are not discussed in detail in the discussion. We have now elaborated some more on the most important results, especially the observations that were reported in more than one study. We added the following sections, which can also be found in the document with tracked changes: “Furthermore, there were significant increases in Actinobacteria, Proteobacteria and Firmicutes in the probiotic groups compared to the control groups. Moreover, the microbiota composition of the experimental group was significantly different from the control group in two studies, bacterial species diversity was increased in one study after administration of a probiotic or synbiotic.” […] “Moreover, three studies on pro- and synbiotics also found a significantly different microbiota composition or a more diverse microbiota in the intervention groups. In general, the microbiome of healthy newborn infants is characterised by low diversity of bacterial species until solid foods are introduced. Birth following Caesarean section or exposure to antibiotics in the first week of life reduce this diversity to such an extent that makes them susceptible to colonisation by bacteria usually found on the mother’s skin such as Staphylococcus, Corynebacterium and Propionibacterium spp., which is associated with an increased risk of allergies later in life.” […] “Three articles based on two prebiotic studies reported an increase in Bifidobacteria and two of the three also found a significantly different microbiota composition”

We also wanted to include in the tables the changes that were found in only one study to show exactly what is known and which findings need more research to draw clear conclusions on the effects of pre-, pro- and synbiotics.

Other observations are:

Abstract

• Selection criteria should be presented in the abstract.

We added the inclusion criteria to the abstract.

• It is important to state the design of the studies selected for the review. 

We added that studies had to be randomised controlled trials to be included in this review.

• Bifidobacterium and Lactobacillus are not species. Please define them or re-write the sentence.

Thank you for this correction. We clarified that the supplemented species were part of the Bifidobacterium and Lactobacillus genus.

• When reporting variations in gut microbial populations, please indicate the group of comparison.

We have now specified in the introduction of the results that the groups of comparisons can be found in Table 1. We chose not to describe each control group in the text of the results to maintain the readability and clarity of this section.

Introduction: 

• Why are authors naming “microbiome” to the “collection of microorganisms”? I am aware that many researchers use the term “gut microbiota” and “gut microbiome” indistinctly, but in essence, these terms don´t mean the same. Please explain.

Thank you for your correction, we changed “microbiome” to “microbiota”.

• The lack of vertical mother-to-infant transmission is not only the possible cause of microbial dysbiosis in the newborn. Please complete.

In this section of the introduction, we specifically aimed to explain how Caesarean section causes microbial dysbiosis. We have now elaborated on the hypothesis that the intrapartum antibiotics during a Caesarean section also increases microbial dysbiosis in the introduction: “It has been suggested that the prenatal antibiotic exposure during a Caesarean section also affects the infant’s microbiota development, but a recent randomised controlled trial (RCT) reported that prenatal exposure to antibiotics during caesarean section does not further disrupt the microbiota colonisation.” [reference: Dierikx, T., Berkhout, D., Eck, A., Tims, S., van Limbergen, J., Visser, D., ... & de Meij, T. (2022). Influence of timing of maternal antibiotic administration during caesarean section on infant microbial colonisation: a randomised controlled trial. Gut, 71(9), 1803-1811.]

• When authors write about antibiotic exposure, they seem to present this as an issue independent of caesarean section. However, they should bear in mind that antibiotic administration to the mother also occurs during caesarean section.

Of course, antibiotic exposure also occurs during caesarean section. One of our co-authors, however, investigated the effect of maternal antibiotics before or after caesarean section on the infant’s microbiota and found no significant differences in the colonisation of the developing microbiota (see the article quoted above). We have clarified this topic in the introduction, as quoted above.

• The definitions of pre, pro and synbiotics should be presented.

We provided definitions of pre-, pro- and synbiotics in the second to last paragraph of the introduction: “Probiotics are live microorganisms such as Bifidobacteria and Lactobacilli,[6] while prebiotics are nutrients that promote growth and activity of bacteria that already exist in the gut.[19] Synbiotics are a combination of pre- and probiotics.[19]”

Results

• Authors should indicate the reasons of the elimination of the articles in the first step of the screening (11193).

We added an explanation of how the title/abstract screening process was executed to the methods section. (“Titles and abstracts were screened by determining whether the article could meet the in- and exclusion criteria stated above.”)

• Table 1: The interventions must be given in detail regarding the CFU of the probiotics (alone or in combination with prebiotics in symbiotic preparations) the bacterial strains used in the probiotics and in the synbiotic formulations, the prebiotic amount provided per dose, the dairy dose, and the administration route.

We agree with the reviewer that these data are important. However, given the fact that Table 1 is already very extensive, we decided not to include this specific information. Interested readers can find these data in the full texts of the included articles.

• Footnote of table one present two times “I”, for “Intervention” and for OUT.

Thank you for your correction, we changed the “I” for operational taxonomic units to “OTU”.

• Critical appraisal: authors should indicate what each color means.

The definitions of each colour were stated in the legend of the table. To make this clearer, we put the colour definitions at the top and made them bold to stand out.

• Please check for mistakes in the bacterial nomenclature of bacteria (specially the use of italics) throughout the manuscript, including the tables.

Thank you for bringing this to our attention, we made sure to correct the use of italics in the bacterial nomenclature.

• Authors state that “this is the first systematic review evaluating the effects of pre-, pro- and synbiotics specifically in both Caesarean born and antibiotic-exposed infants”. However only 1 out of the 12 studies are actually dealing with antibiotics.

While we agree that it is a shame that there is only one study investigating the effect of pre-, pro- and synbiotics on infants exposed to antibiotics in early life, we did thoroughly look for any articles dealing with this topic. Furthermore, while we identified one other review on the effects of pre-, pro- and synbiotics on Caesarean-born infants, they also included studies that included vaginally born infants as well without performing subgroup analyses. Therefore we feel justified in saying that this is the first review evaluating the effects of pre-, pro- and synbiotics specifically in Caesarean-born and antibiotic-exposed infants.

Reviewer #3:

Dear Authors,

The manuscript is in interesting filed. However, there are some major concerns.

Overall, whole manuscript is required to English language editing.

We asked a native speaker review and correct the manuscript.

Then, please consider following comments.

Title:

-It is better to brief.

We removed the unnecessary word “supplemented” from the title.

Abstract:

-Methods section: Please insert inclusion/exclusion criteria, main keywords, update the search, and insert grey literature

We inserted the inclusion/exclusion criteria and performed a new search, after which we included one more article by Estorninos et al. Furthermore, we aimed to summarise all relevant published articles but no unpublished papers (grey literature) as it would not be possible to search systematically for such literature. We recently repeated our search, so we believe that any studies described in trial registers for example that have now published their manuscripts would have come up in this second search.

Introduction:

-Nearly, more than half of Ref. are addressed in this section, please reduce Ref.

We removed some unnecessary double citations and reduced the number of references in the introduction from 28 to 18.

-Please address past systematic reviews and then the novelty of the current study.

We address the only other similar review we found in the discussion section and also explain why we think our review is an improvement: “While another systematic review about the effects of pre-, pro- and synbiotics on the microbiota of children born via Caesarean section was published recently,[43] we identified four additional relevant articles that were not included by Martin-Pelaez et al. Furthermore, some of their included articles did not perform a separate subgroup-analysis for children born via Caesarean section.”

Methods:

-It is better to update search, insert grey literature, and confirm the method is based on PRISMA.

We updated our search on August 10 2022 and included one more article. We added that we wrote the review according to the PRISMA statement to the methods section.

-Did you search ref. lists? It is essential to hand search Ref. list of the included studies when conducting a systematic review.

We hand searched the reference lists of the included articles and added this to the methods section.

-In eligibility criteria, clarify characteristics of control group, they should receive which one? prebiotics, placebo, or none of them?

When we composed the in- and exclusion criteria before starting our search, we did not choose to limit our search to specific control groups because we aimed to summarise all available evidence on the effect of pre-, pro- and synbiotics on the microbiota of infants born via Caesarean section or exposed to antibiotics in the first week of life.

-The second inclusion criteria “six weeks after birth” is ambiguous. In table 1 is observed using intervention more than 6w. What is your reason?

We have now clarified in the methods section that administration of the intervention should be started within 6 weeks after birth: “administration of pre-, pro- or synbiotic dietary supplements was started within six weeks after birth,”. We had to decide on a cut-off point before starting the title/abstract screening and we expected that, if interventions were administrated after >6 weeks, there would be too many other factors affecting the infants’ microbiota.

-In critical appraisal; the last sentence “Several studies included…..”should move to discussion.

We changed this sentence to: “If a study included both vaginally and Caesarean-delivered infants and performed a subgroup analysis on the Caesarean-delivered infants, only the methods used for the relevant subgroup analyses were assessed” to make it suitable for the methods section.

-Please delete “the review and protocol were not registrated”.

This sentence cannot be deleted because the PRISMA checklist asks authors to include this sentence in point 24a.

Results:

-First line: correct dictation “deduplication”.

We believe that “deduplication” is the correct term for removing copies of data.

-“ Lay et al.[33] published results of a subgroup analysis based on the RCT by Chua et al.[34]” Why included this study? It is a duplication publication. Please describe.

We included both studies because Lay et al. performed additional analyses on a subgroup of the infants of Chua et al. Therefore, they have some different results from Chua et al. and we did not state in our in- and exclusion criteria that we would exclude such articles. However, we tried to make it very clear throughout the paper that these articles were based on the same study to limit any confusion caused by the inclusion of both papers.

-Zhong et al study is included vaginal/cesarean delivery. Please recheck the data with original paper and correct them in your tables.

Zhong et al. is the only study in our review that investigated infants who were exposed to antibiotics in the first week of life instead of infants born following caesarean section, so in this study vaginally born infants were also included. We clarified this in our inclusion criteria: “(1) study participants were term-born infants who were born following Caesarean section or exposed to antibiotics in the first week of life (born vaginally or following Caesarean section)”. Furthermore, we separated this study from the other studies in tables 1 and 2 by putting a bold line in between the study of Zhong et al. and the other studies focusing on Caesarean born infants.

Discussion:

-This section needs to revise by adding more details and discussion about your results.

We elaborated on the results found in more than one study and also explained the possible clinical relevance: “Furthermore, there were significant increases in Actinobacteria, Proteobacteria and Firmicutes in the probiotic groups compared to the control groups. Moreover, the microbiota composition of the experimental group was significantly different from the control group in two studies, bacterial species diversity was increased in one study after administration of a probiotic or synbiotic.” […] “Moreover, three studies on pro- and synbiotics also found a significantly different microbiota composition or a more diverse microbiota in the intervention groups. In general, the microbiome of healthy newborn infants is characterised by low diversity of bacterial species until solid foods are introduced. Birth following Caesarean section or exposure to antibiotics in the first week of life reduce this diversity to such an extent that makes them susceptible to colonisation by bacteria usually found on the mother’s skin such as Staphylococcus, Corynebacterium and Propionibacterium spp., which is associated with an increased risk of allergies later in life.” […] “Three articles based on two prebiotic studies reported an increase in Bifidobacteria and two of the three also found a significantly different microbiota composition”

---

## [Decision Letter · Decision Letter 1]

4 Oct 2022

PONE-D-22-18775R1Microbial effects of prebiotics, probiotics and synbiotics after Caesarean section or exposure to antibiotics in the first week of life: a systematic reviewPLOS ONE

Dear Dr. Carpay,

Thank you for submitting your manuscript to PLOS ONE. After careful consideration, we feel that it has merit but does not fully meet PLOS ONE’s publication criteria as it currently stands. Therefore, we invite you to submit a revised version of the manuscript that addresses the points raised during the review process.

We look forward to receiving your revised manuscript.

Kind regards,

Ozra Tabatabaei-Malazy

Academic Editor

PLOS ONE

Reviewers' comments:

Reviewer's Responses to Questions

**Comments to the Author**

1. If the authors have adequately addressed your comments raised in a previous round of review and you feel that this manuscript is now acceptable for publication, you may indicate that here to bypass the “Comments to the Author” section, enter your conflict of interest statement in the “Confidential to Editor” section, and submit your "Accept" recommendation.

Reviewer #3: (No Response)

Reviewer #4: (No Response)

2. Is the manuscript technically sound, and do the data support the conclusions?

Reviewer #3: Partly

Reviewer #4: Yes

3. Has the statistical analysis been performed appropriately and rigorously? 

Reviewer #3: Yes

Reviewer #4: N/A

4. Have the authors made all data underlying the findings in their manuscript fully available?

Reviewer #3: Yes

Reviewer #4: Yes

5. Is the manuscript presented in an intelligible fashion and written in standard English?

Reviewer #3: Yes

Reviewer #4: Yes

6. Review Comments to the Author

Reviewer #3: Dear Authors,

The quality of the manuscript is improved. However, it should be considered following comments.

-Considering clinical trials as one of the inclusion criteria, why is not limited the search strategy to it? It could reduce the initial records, and time of assessment.

-Please insert type of trials in table-1.

-According to following evidence, mode of delivery has influenced on gut microbiota of the infants. I suggest to exclude Zhong study and also revised the title and inclusion criteria.

The mode of delivery affects the diversity and colonization pattern of the gut microbiota during the first year of infants' life: a systematic review. BMC Gastroenterol. 2016 Jul 30;16(1):86. doi: 10.1186/s12876-016-0498-0.

Best Regards,

Reviewer #4: Carpay and colleagues provided a summary of the research on the impact of prebiotics, probiotics, and symbiotics on the makeup of the intestinal microbiota in term infants who were delivered by Caesarean section or who had received antibiotics during the first week of life. The patient, group of patients, or issue being examined; the intervention; comparator interventions; and particular outcomes are the aspects that the authors have addressed that are essential for a systematic review. The rational of the systematic review appears reasonable for me and the authors synthesized the results in a very informative and efficient way. The purpose and backgrounds are appropriately addressed in the introduction and discussion section. Methods section was written at its best. Tables and figures were adequate and inform the readers all the necessary information. Whenever it is questioned, the authors offer the necessary clarification in the discussion area. The review's limitations as well as the ones of the included research have been discussed by the authors. The authors have assessed the potential for bias and taken into account probable RoBs. The manuscript as it is now acceptable for publication in my opinion.

7. PLOS authors have the option to publish the peer review history of their article (what does this mean?). If published, this will include your full peer review and any attached files.

Reviewer #3: No

Reviewer #4: No

---

## [Author Response · Author response to Decision Letter 1]

23 Oct 2022

Amsterdam, October 2022

PLOS ONE

Dr. Tabatabaei-Malazy

Dear Dr. Tabatabaei-Malazy,

Thank you for the opportunity to revise our manuscript “Microbial effects of prebiotics, probiotics and synbiotics after Caesarean section or exposure to antibiotics in the first week of life: a systematic review”. 

We thank the reviewers for taking the time to critically review and comment upon our manuscript. Our response to the reviewers’ comments can be found below. We hope this clarifies any ambiguities and that the revised manuscript meets the standards for publication in PLOS ONE.

We look forward to hearing from you regarding our submission, and we would be happy to address any further comments.

With kind regards,

on behalf of all authors,

Nora Carpay

Reviewer #3

Dear Authors,

The quality of the manuscript is improved. However, it should be considered following comments.

- Considering clinical trials as one of the inclusion criteria, why is not limited the search strategy to it? It could reduce the initial records, and time of assessment.

In hindsight, this would have been a good option. However, we wanted to make sure that we did not omit important studies of other study designs which we could include in our discussion. In the end, we did not find any relevant studies of other study designs.

- Please insert type of trials in table-1.

Thank you for your suggestion. To maintain the readability of table 1, we changed the title to “General characteristics of the included randomised controlled trials” instead of “General characteristics of the included studies” to clarify.

- According to following evidence, mode of delivery has influenced on gut microbiota of the infants. I suggest to exclude Zhong study and also revised the title and inclusion criteria.

The mode of delivery affects the diversity and colonization pattern of the gut microbiota during the first year of infants' life: a systematic review. BMC Gastroenterol. 2016 Jul 30;16(1):86. doi: 10.1186/s12876-016-0498-0.

We agree that mode of delivery affects the gut microbiota of infants. However, so does the administration of antibiotics in the first week of life*. In this review we therefore aimed to investigate the effect of pre-, pro- and synbiotics on the gut microbiota of infants born via C-section or after antibiotic exposure. Unfortunately, we only found one study (by Zhong et al.) about the effects of such supplements on infants exposed to antibiotics, but since this was our original question, we do think it is important to include this study to illustrate the need for further research. Furthermore, we do not want to change our initial aim, as this would lead to reporting bias.

*Van Daele E, Kamphorst K, Vlieger AM, Hermes G, Milani C, Ventura M, Belzer C, Smidt H, van Elburg RM, Knol J. Effect of antibiotics in the first week of life on faecal microbiota development. Arch Dis Child Fetal Neonatal Ed. 2022 May 9:fetalneonatal-2021-322861. 

Reviewer #4

Carpay and colleagues provided a summary of the research on the impact of prebiotics, probiotics, and symbiotics on the makeup of the intestinal microbiota in term infants who were delivered by Caesarean section or who had received antibiotics during the first week of life. The patient, group of patients, or issue being examined; the intervention; comparator interventions; and particular outcomes are the aspects that the authors have addressed that are essential for a systematic review. The rational of the systematic review appears reasonable for me and the authors synthesized the results in a very informative and efficient way. The purpose and backgrounds are appropriately addressed in the introduction and discussion section. Methods section was written at its best. Tables and figures were adequate and inform the readers all the necessary information. Whenever it is questioned, the authors offer the necessary clarification in the discussion area. The review's limitations as well as the ones of the included research have been discussed by the authors. The authors have assessed the potential for bias and taken into account probable RoBs. The manuscript as it is now acceptable for publication in my opinion.

We thank the reviewer for taking the time to review our manuscript, and for their positive feedback.

---

## [Editor Report · Decision Letter 2]

26 Oct 2022

Microbial effects of prebiotics, probiotics and synbiotics after Caesarean section or exposure to antibiotics in the first week of life: a systematic review

PONE-D-22-18775R2

Dear Dr. Carpay,

We’re pleased to inform you that your manuscript has been judged scientifically suitable for publication and will be formally accepted for publication once it meets all outstanding technical requirements.

Kind regards,

Ozra Tabatabaei-Malazy

Academic Editor

PLOS ONE